# De Novo Transcription Responses Describe Host-Related Differentiation of *Paracoccus marginatus* (Hemiptera: Pseudococcidae)

**DOI:** 10.3390/insects13090850

**Published:** 2022-09-19

**Authors:** Lizhen Zheng, Jianyu Li, Mengzhu Shi, Yanting Chen, Xiaoyun He, Jianwei Fu

**Affiliations:** 1Institute of Plant Protection, Fujian Academy of Agricultural Sciences, Fuzhou 350003, China; 2Fujian Provincial Key Laboratory of Quality and Safety of Agricultural Products, Institute of Quality Standards & Testing Technology for Agro-Products, Fujian Academy of Agricultural Sciences, Fuzhou 350003, China

**Keywords:** invasive pests, host-related differentiation, transcription

## Abstract

**Simple Summary:**

The papaya mealybug, *Paracoccus marginatus*, is an invasive pest affecting many crop plants. It reproduces and spreads rapidly. They have historically been a pest of potato, *Solanum tuberosum,* but they have successfully adapted to infesting papaya, *Carica papaya*. When they feed on papaya, they survive and reproduce more and live longer than when they feed on potato. We do not yet know what biological adaptations they made in order to use this new host plant. We compared the RNA sequences of papaya mealybugs feeding on potato and papaya. A total of 408 genes are expressed differently depending on the host plant. Most of these genes are expressed less when feeding on potato than on papaya. They encode digestive enzymes, detoxifying enzymes, and ribosomes and some have reproductive functions. We further analyzed their known functions using the Kyoto Encyclopedia of Genes and Genomes. This showed that they include genes regulating digestion, detoxification, and longevity. We suggest that papaya is a more suitable host than potato, and that the decreased expression of particular genes may have important effects on the adaptation of the papaya mealybug to this alternative host plant.

**Abstract:**

*Paracoccus marginatus* (Hemiptera: Pseudococcidae) is an invasive pest with a diverse host range, strong diffusion, and high fecundity. It has been observed that *P. marginatus* feeding on *Carica papaya* have a higher survival rate, fecundity, and longer lifespan than *P. marginatus* feeding on *Solanum tuberosum*, indicating their successful adaptation to *C. papaya*; however, the mechanisms underlying host plant adaptation remain unclear. Therefore, RNA-seq was performed to study the transcriptional responses of *P. marginatus* feeding on *C. papaya* and *S. tuberosum* plants. A total of 408 genes with significant differential expression were defined; most of them were downregulated in *S. tuberosum*, including those of digestive enzymes, detoxifying enzymes, ribosomes, and reproductive-related genes, which may result from the adaptation of the host to nutritional needs and changes in toxic chemical levels. Enrichment analysis of the Kyoto Encyclopedia of Genes and Genomes showed that lysosome and longevity regulating pathways related to digestion, detoxification, and longevity were enriched. We suggest that *C. papaya* is a more suitable host than *S. tuberosum*, and downregulated target genes may have important effects on the adaptation of *P. marginatus* to host transfer.

## 1. Introduction

Coevolution between host plants and phytophagous insects is a frequent phenomenon always in a competitive state [1]. Plants have evolved various methods to reduce consumption by phytophagous insects, while these insects have developed defensive mechanisms to deal with their host plants or find new hosts. Plants have evolved physical barriers to inhibit insect colonization [2]. They synthesize toxic compounds (including allelochemicals), and different nutritional levels may reduce the growth, survival, and fecundity of insects [3,4,5]. Insects develop complex defense systems, including allelochemical transformation and excretion, to adapt to host plants and overcome their defenses so that they can survive on specific plant species or find new hosts [6,7]. To better understand these mechanisms, some studies have explored the transcriptional responses of insect host plants to host transfer [1].

*Paracoccus marginatus* (Hemiptera: Pseudococcidae) is an invasive pest with a diverse host range, strong diffusion, and high fecundity. It takes about 30 days to complete a generation [8]. It was first reported to be harmful in Saint Martin in 1995, and since then, has spread to at least 40 countries and caused substantial economic losses to the *C. papaya* industry in Central America, North America, the Pacific, Africa, and Asia [9,10,11]. *Paracoccus marginatus* is an aggressive pest with a wide host range of 68 families (264 species), including fruits, grains, ornamental flowers, and weeds [12]. Similar to other mealybugs, *P. marginatus* absorbs plant juice by inserting its stylet into the epidermis of fruits, leaves, and stems. As a result, leaves curl, wrinkle, rosette, twist, and become generally distorted. The dense *P. marginatus* population excretes large quantities of honeydew. The white wax accumulated during the growth process can induce sooty mold, which ultimately affects the fruit’s edibility and market value [13,14].

*Paracoccus marginatus* feeding on *C. papaya* has a shorter developmental time, longer lifespan, and higher pre-adult survival rates and fecundity than those feeding on *S. tuberosum*, implying that *C. papaya* is more suitable for the growth and survival of *P. marginatus* than *S. tuberosum* [15]. This indicates that the biochemical compositions and secondary metabolic products of these hosts may differ from each other [16]. However, the molecular mechanisms of host plant adaptations by *P. marginatus* remain unknown. In this study, transcriptional analysis of *P. marginatus* offers a theoretical basis for future research at the molecular level.

The overall analysis of the transcriptome response is an effective method for understanding the molecular mechanism of insect adaptation to plant defense [2,17,18,19,20]. Previous work has mainly studied herbivorous insects hosted by different plants, detecting expression patterns of different genes, including those related to digestion, detoxification, ribosomes, and reproduction-related genes [21,22,23,24,25]. We know little about the transcriptomic response of herbivorous insects after a host shift [26]; there are few studies on the molecular mechanism of this process. Transcriptional analysis of *P. marginatus* feeding on *C. papaya* and *S. tuberosum* was used in this study to identify differentially expressed genes correlating to host adaptation. This study is the first to explore transcriptional changes in *P. marginatus* as it adapts to different hosts.

## 2. Materials and Methods

### 2.1. Specimen Rearing and Collection

In 2018, *P. marginatus* adults were collected from *C. papaya* trees west of the Fujian Agriculture and Forestry University, Fujian, China. They were reared in an incubator under the following conditions: temperature of 28 ± 1 °C, relative humidity of 70 ± 5%, a photoperiod of 12 h light and 12 h dark, and light intensity of 12,000 lx. The subculture’s host was germinated *S. tuberosum*, the insects were preserved for several generations, and then transfer to two different undamaged hosts. To test the developmental stages, fecundity, and survival rate of the *P. marginatus* on two host plants, we refitted a 500 mL transparent plastic box (height 10 cm, diameter 12 cm), cut out a round hole with a diameter of approximately 6 cm, and covered it with 200-µm mesh nylon mesh (Appendix A). For 100 eggs from the test hosts *S. tuberosum* and *C. papaya*, the hatching, growth and development, survival, and death rates were recorded every day. After mating, the daily fecundity and survival rates were recorded until death. The experimental data were analyzed using the paired bootstrap in TWOSEX-MSChart software [27].

To analyze the molecular mechanism of the effects of the two hosts on the growth and reproduction of *P. marginatus,* we collected adult females of each generation from generation 0 to generation 6. Each generation took approximately a month to rear. RNA-seq analyses were performed on samples of 20 adults of *P. marginatus* from each host, collected at the end of the sixth generation, the *C. papaya* sixth generation (MF6) was used as a control, and the *S. tuberosum* sixth generation (TF6) as the treatment. Three independent duplicate samples for each control and treatment were taken (MF6_1, MF6_2, and MF6_3; TF6_1, TF6_2, and TF6_3), and they were quickly stored in liquid nitrogen and further at −80 °C for standby.

### 2.2. RNA Isolation and Transcriptome Sequencing

TRIzol reagent (Invitrogen, Carlsbad, CA, USA) was used to isolate the total RNA as per the manufacturer’s protocol, and a NanoPhotometer^®^ spectrophotometer (IMPLEN, Westlake Village, CA, USA) was used to analyze the purified total RNA samples. The RNA quality was assessed using an Agilent 2100 system (Thermo Fisher Scientific, Waltham, MA, USA). High-quality RNA was extracted from three biological repeat samples for each *P. marginatus* population to construct cDNA libraries and then sequenced on the BGISEQ500 platform (BGI Co. Ltd., Shenzhen, China). Finally, 100-bp paired-end raw reads were produced. All raw read sequences were stored under the accession number PRJNA769538 of the NCBI Short Read Archive (SRA) database.

### 2.3. Bioinformatics Analysis

By removing the reads that included poly-N adaptors or low-quality ones, the raw reads were filtered out via SOAPnuke (v1.4.0) [28] and clean reads were obtained, which were subsequently assembled using Trinity (v2.0.6) [29]. To obtain the unique genes, Tgicl (v2.0.6) [30] was used to cluster and remove redundant data in the compiled transcripts.

Expression analysis and functional annotation were performed on the assembled transcripts. Clean readings were mapped to the assembled unique genes using bowtie2 (v2.2.5) [31], and RSEM (v1.2.8) [32] was used to measure the gene expression levels and normalize the fragments per kilobase of transcript per million mapped reads (FPKM). Gene function annotation was realized using the basic local alignment search tool (BLAST, v2.2.23) [33] by mapping the genes to different databases (protein (nr), nucleotide (nt), EuKaryotic Orthologous Groups (KOG), and Kyoto Encyclopedia of Genes and Genomes (KEGG)). Gene Ontology (GO) annotation was performed using Blast2GO (v2.5.0) with NR annotations, and DEseq2 [34] was used to find differentially expressed genes (DEGs) with a fold change of >2 or <−2; significantly DEGs were adjusted by a *p*-value of ≤0.001. Phyper (a function of R) was used to analyze the GO enrichment and KEGG enrichment, and a strict threshold (Q value of <0.05) was used to define the significance levels of terms and pathways.

### 2.4. qRT-PCR Analysis

To confirm the transcriptome results, 10 DEGs with high expression levels in the enrichment pathway were selected from the *C. papaya* and *S. tuberosum* treatments for qRT-PCR. In addition to transcriptome sequencing, the HiScript Q RT SuperMix for the qPCR (+gDNA wiper) Kit (Vazyme Biotech Co., Ltd., Nanjing, China) was used to synthesize the cDNA as per the instructions, and on the ABI7500 fluorescence quantitative PCR instrument (Applied Biosystems, Waltham, MA, USA) with designed primers (Appendix A). The tubulin gene was used for transcript normalization. The total reaction volume for the amplification was 20 μL, including 10 μL of 2× ChamQ SYBR Color qPCR Master Mix (Vazyme), 0.8 μL of each primer (5 μM), 2 μL of cDNA,0.4 μL of 50× ROX Reference Dye 2, and 6 μL of ddH_2_O. The qPCR thermal cycling was performed at 95 °C for 5 min, followed by 40 cycles at 95 °C for 5 s, 55 °C for 30 s, and 72 °C for 40 s. Using the 2^−ΔΔCT^ method [35], the relative expressions of 10 genes were calculated in the *C. papaya* and *S. tuberosum* treatments of *P. marginatus*. The relevance among multiple changes of qRT-PCR and RNA-seq gene expression rates was analyzed using Student’s *t*-test and Origin 2017 (http://www.originlab.com/, accessed on 24 October 2016).

## 3. Results

### 3.1. Development Duration, Survival Rate, and Fecundity of P. marginatus on Two Host Plants

The development time of the *P. marginatus* raised on *C. papaya* was shorter for the third instar, pupa, and male adult (Table 1), but the female adult stage was significantly longer on *C. papaya* compared to *S. tuberosum* (*p* < 0.001). There were no significant differences in the other development times. The longevity on the two hosts was similar for the adult males but significantly longer for the adult females on *C. papaya* than on *S. tuberosum*. In general, *P. marginatus* reared on *C. papaya* had faster development and a longer lifespan, indicating that *C. papaya* is a more suitable host plant for *P. marginatus* than *S. tuberosum*.

The fecundity of *P. marginatus* was significantly higher on *C. papaya* than on *S. tuberosum* (*p* < 0.001) (Table 2). The oviposition period on *C. papaya* was also significantly longer than that on *S. tuberosum*. The pre-adult survival of *P. marginatus* on *C. papaya* was significantly higher than that on *S. tuberosum* (*p* = 0.001), indicating that *S. tuberosum* was more unfavorable to the growth and development of *P. marginatus* nymphs than *C. papaya*. There were no significant differences in the sex ratio or fertile female ratio between *C. papaya* and *S. tuberosum*. 

### 3.2. Assembled Transcriptome

The results of the high-throughput sequencing showed that the average reading capacity of the six transcriptional libraries from the *P. marginatus* groups reared on *C. papaya* (MF6_1, MF6_2, and MF6_3) and those raised on *S. tuberosum* (TF6_1, TF6_2, and TF6_3) was 73.05 M (Table 3). We obtained a clear reading of 66–70 M, including 6–7 billion nucleotides (6–7 GB) after data filtering, and each sample was more than 6.69 GB, with an average length greater than 1425 bp and N50 more than 2274 bp.

The average GC content was 39.54%. The percentage of the Q20 base was more than 97.41%, and the high sequencing quality with a Q30 ratio of all the samples was greater than 89%. There were 19,455 (62.41%) unigenes over 1000 bp; the length distributions of all unigenes are shown in Appendix A.

### 3.3. Functional Annotation

From the high-quality reads, 31,175 unigenes were generated by querying seven databases for accurate annotation (Table 4).

Furthermore, the unigenes of *P. marginatus* were searched for classification and functional prediction in the KOG database. 17,508 genes in total (56.16% of all the unigenes) were aligned to the KOG classification (Table 4, Figure 1) and divided into 25 different categories (Appendix A). The maximal category was “general function prediction only” (3495 unigenes, 19.96%), followed by “signal transduction mechanisms” (2072 unigenes, 11.83%), “posttranslational modification, protein turnover, chaperones” (1334 unigenes, 7.62%), “function unknown” (1171 unigenes, 6.69%), and “transcription” (1115 unigenes, 6.37%). Only some unigenes were aligned to “cell motility” (25 unigenes, 0.16%), the smallest group.

### 3.4. Principal Component Analysis (PCA)

RNA-seq data from all samples (MF6 and TF6) were replicated three times. The replicates of the three samples were relatively more closely related than the other samples (Figure 2). MF6_3 was located close to MF6_1 and MF6_2, but further from TF6_1, TF6_2, and TF6_3, indicating greater gene expression similarity in the MF6 and TF6 groups.

### 3.5. Analysis of DEGs

Compared to the control (MF6), the expressions of the 4400 TF6 genes differed significantly (*p*-value < 0.05), with 2394 genes downregulated and 2006 genes upregulated (Figure 3, Appendix A).

Using Blast2GO, and as per the annotation results of Nr and Pfam, GO annotation was performed to classify the functions of all DEGs. As shown in Figure 4, the unigenes were divided into 49 terms, including 16 (32.65%) cellular component terms, 22 (44.90%) biological process terms, and 11 (22.45%) molecular function terms. In the cellular component category, cell (390) and cell part terms (378) were the most abundant terms, followed by membrane (360). For the molecular function category, binding (525) was the predominant term, followed by catalytic activity (468) and structural molecule activity (85). Within the biological process category, cellular process (418) and metabolic process (388) were the most abundant terms, followed by biological regulation (144), indicating that the host’s feeding and absorption are mainly reflected in the metabolic pathway and cellular processes.

To identify the biological pathways in the transcriptome of *P. marginatus*, unigenes were mapped to the KEGG database for enrichment analysis using the Phyper function in R. A total of 1718 unigenes were divided into 242 KEGG pathways (removing human diseases, Appendix A). For all genes involved in the pathway, metabolism was the largest category (1137/3695, 30.77%), followed by organismal systems (1113/3695, 30.12%), cellular processes (585/3695, 15.83%), environmental information processing (495/3695, 13.40%), and genetic information processing (365/3695, 9.88%, Figure 5). Similar to the GO classification, these findings show that metabolic processes were active in *P. marginatus,* which shows that this species synthesizes various metabolites.

### 3.6. Comparative Analysis of Transcriptome

To obtain the molecular mechanisms of the host adaptability in *P. marginatus*, we compared and analyzed the six transcriptomes. Using a *p*-value of <0.05, fold change values of >2, and an FPKM of >1, we identified 408 DEGs in TF6 compared to MF6, including 145 upregulated genes and 263 downregulated genes, primarily participating in physiologic functions, including detoxification, reproduction, digestion, and metabolism (Appendix A). Common detoxification-related DEGs included esterase, UGTs, and P450s; most of these were downregulated. The gene expression pattern between MF6 and TF6, presented in Figure 6, illustrates that most genes were adjusted when adapting to different plants.

Using Blast2GO, GO enrichment analysis was conducted to understand the possible functions of DEGs. The GO functional classification results of the three main GO domains could be classified into 35 functional groups: cellular components (10), molecular functions (8), and biological processes (17). Within these groups, we found metabolic processes (86 DEGs) and cellular processes (74 DEGs) in the biological process ontology, catalytic activity (102 DEGs), and binding (70 DEGs) in the molecular function ontology, and cell (64 DEGs) and cell part (60 DEGs) were the dominant annotation terms in the cellular component ontology (Appendix A). Enriched GO terms are mainly involved in detoxification, digestion, and metabolism-related functions (Appendix A). Most DEGs were downregulated in GO terms (Figure 7), implying that most of the downregulated DEGs might be associated with host adaptation in *P. marginatus*.

Using KOBAS 2.0, KEGG pathway analysis was performed to better understand the functions of DEGs. The results show that DEGs were involved in 207 KEGG pathways (removing human diseases). The largest unigenes group of the top 50 KEGG pathways is shown in Appendix A. The most dominant pathways were “Steroid hormone biosynthesis” (ko00140), “Glyoxylate and dicarboxylate metabolism” (ko00630), and “Glycerolipid metabolism” (ko00561) (Figure 8). Detoxification, digestion, metabolism, and growth pathways are all displayed. The detoxification pathways that were significantly enriched primarily contained “drug metabolism cytochrome P450”, while the digestion pathways that were significantly enriched primarily contained “fat digestion and absorption” and “lysosome”. We then implemented an enrichment analysis of KEGG terms for the upregulated and downregulated DEGs. Similar pathways were reflected in the downregulated DEGs (Appendix A). The enriched pathways differed among the upregulated DEGs, including “longevity regulating pathway-worm” (Appendix A), suggesting that this pathway may have an important role in the lifespan of *P. marginatus* when fed on *S. tuberosum*.

According to the KEGG enrichment analysis, 207 pathways were significantly enriched (Appendix A). The lysosome pathway was the most prevalent KEGG pathway. Twenty-three DEGs related to self-renewal tissue and digestion were observed in the “lysosome” pathway. Among these, three genes were upregulated in TF6 compared to MF6, including cystinosin (CL1098.Contig5_All), AP-3 (CL3350.Contig2_All), and LAMAN (CL626.Contig11_All). Twenty genes were downregulated, including cathepsins (CL1063.Contig2_All, CL3096.Contig1_All, CL3981.Contig2_All, Unigene11804_All and CL2404.Contig1_All), AP-1 (CL1124.Contig2_All and CL1124.Contig3_All), LIPA (CL2194.Contig1_All and CL2667.Contig4_All), LAMAN (CL286.Contig13_All, Unigene10749_All, Unigene8232_All, and CL286.Contig8_All), LGMN (CL2961.Contig1_All and CL2961.Contig2_All), ARS (CL4.Contig2_All), GAA (Unigene10628_All), NPC (Unigene10809_All), LGMN (Unigene11195_All), and sialin (Unigene149_All) (Figure 9). In the “longevity regulating pathway-worm” pathway, six DEGs related to longevity were upregulated in TF6 compared to MF6. These included DAF-18 (CL525.Contig3_All), FARD-1 (CL1378.Contig3_All and CL2010.Contig1_All), and FAT-6 (CL196.Contig2_All, CL196.Contig3_All, and CL2040.Contig1_All) (Appendix A).

Four classes of DEGs that function as metabolic and reproduction-related genes were identified: digestive enzyme genes, detoxifying enzyme genes, reproduction-related genes, and ribosomes. DEG analysis revealed 32 digestive enzyme unigenes, including 26 downregulated and six upregulated unigenes. Twenty-two detoxifying enzyme genes were differentially regulated, including 17 downregulated and five upregulated unigenes. These included cytochrome P450, esterase, peroxidase, and UGT genes. Six reproduction-related genes were downregulated, and none were upregulated. Five ribosome protein genes were detected, including three downregulated and two upregulated unigenes. Accordingly, most of the target genes were downregulated and few were upregulated (Table 5 and Appendix A in detail).

To validate the transcriptional comparison results, we selected 10 unigenes from the targeted genes for qRT-PCR verification. The qRT-PCR expression profiles of the candidate unigenes were similar to those of transcriptome DEGs. The expression level of tubulin was stable enough to be used as an internal control (Figure 10). Linear regression analysis revealed that the gene expression ratios between qRT-PCR and RNA-seq (R^2^ = 0.48) had a positive correlation (Appendix A), confirming the validity of our transcriptomic data.

## 4. Discussion

Because *P. marginatus* males had similar response patterns in both hosts (see Section 3.1), females were used for the transcriptome experiments. Since genomic information for *P. marginatus* was not available, and the mechanisms of the molecular response of the species to different hosts are not known, we further investigated the host molecular responses of *P. marginatus* to *C. papaya* and *S. tuberosum* using de novo transcriptome sequencing. In the current study, the transcriptional characterization of *P. marginatus* produced more than 66 M clean reads assembled into 31,175 unigenes with more than 1425 bp. Using seven major databases, about 22,973 (73.69%) unigenes were annotated as having biological functions, and 8202 (26.31%) unigenes had no apparent homologs (Table 4). This suggests that *P. marginatus* may have some species-exclusive unigenes.

Insects adapt to host plants and improve their adaptability and resistance by increasing the activity of digestive enzymes in vivo. Digestive enzymes are required for insects to digest food to obtain nutrients, including proteases (serine protease, trypsin, aminopeptidase, cysteine protease, and carboxypeptidase), lipases, and amylases [36]. Host plants with varying nutritional values may require herbivores to express a variety of digestive enzyme activities [20,37,38]. We predicted that the digestive enzyme genes of *P. marginatus* might respond to different plant hosts. In our study, 32 digestive enzyme genes were differentially expressed in *P. marginatus* (Table 5), with 26 downregulated and six upregulated. This shows that the digestive tract of *P. marginatus* is greatly influenced by the host *S. tuberosum*. The majority of DEGs participate in metabolic processes associated with serine protease and lipase and promote the digestion, absorption, transportation, and metabolism of lipids and lipoproteins [39], as well as insect disease resistance and the mediation of plant defense responses [40,41]. One cysteine protease gene was previously upregulated in the tarnished plant bug, *Lygus lineolaris* [42]. However, in our study, four DEGs from the cysteine protease family were significantly downregulated in *P. marginatus* in TF6 compared to MF6. This indicates that the expressions of cysteine protease genes are diverse in responses to host transfer. Another digestive gene was similar to the serine protease genes [43]. Serine proteases are a large gene family in insects, and their main functions include food metabolism, immune defense, and enzyme source activation [44]. Earlier studies have shown that serine proteases constitute the largest gene family in *Drosophila* [45] and are important enzymes used by larvae to utilize different host plants [22,44], which is consistent with our findings. We identified 11 serine proteases previously identified as key enzymes in host adaptation [20,22,44] and studied their expression patterns in different hosts. We also found that the expressions of five trypsin genes were downregulated. Trypsin is a major proteolytic enzyme in insects that can rapidly activate other proteases to perform digestive functions [46]. When fed *S. tuberosum*, most of the digestive enzyme genes in *P. marginatus* were downregulated, indicating that they required less energy for metabolism, development, growth, and reproduction, as was the participation of other proteasogens in digestion and metabolism. The growth and reproductive ability of *P. marginatus* were lower in TF6 (Table 1 and Table 2) [15], suggesting that digestive enzyme genes may play key roles for different hosts in *P. marginatus*.

Phytophagous insects also need to deal with the harmful effects of secondary toxic substances produced by plant defense responses, including xenobiotic excretion, chemical conversion, and reduced absorption of ingested chemosensory substances [3]. The detoxification ability of these compounds can dictate their host scope, so detoxification systems are important for host fitness [20]. Insect detoxifying enzymes, for example, UGTs, P450s, and esterases are involved in endogenous compound metabolism, insecticide resistance, and tolerance to phytotoxic compounds [3,47,48]. Therefore, P450s, UGTs, and esterases are widely regarded as the main detoxifiers of allelochemicals [3,47,49]. The process of detoxification is typically divided into three stages, each involving different detoxifying enzymes [19,22,24,25,38]. Phase I enzymes participate in the routine detoxification of allelochemicals, comprising P450 proteins and esterases [20]; phase II enzymes include UDP-glycosyltransferases (UGTs) [50]. UGTs play important roles in detoxification [51]. RNA-seq is an effective method to explain detoxification system functions related to host adaptation [2,22,24,26]. We screened all esterases, P450s, POD, and UGT genes in the transcriptome sequencing results and found 22 DEGs of detoxifying enzyme genes (Table 5), including 17 downregulated genes and five upregulated genes. Most phase I enzymes (esterase and P450s) and phase II enzymes (UGTs) were downregulated in TF6 compared to MF6, indicating that these genes may participate in the detoxification process in different host plants. After feeding on *S. tuberosum*, the detoxification ability of *P. marginatus* for secondary substances and toxic substances was greatly reduced, so the detoxification capability of TF6 is much lower than that of MF6. Similar results have been found in other insects; detoxification enzyme activity was downregulated when other insects fed on unsuitable hosts [20,49,52,53,54,55]. Moreover, KEGG enrichment showed that 12 DEGs were enriched in “drug metabolism—cytochrome P450” pathways in MF6 vs. TF6, including two upregulated genes and ten downregulated genes, showing that the detoxification metabolism of TF6 was clearly inhibited (Appendix A).

Allelochemicals are major plant defenses; therefore, they serve as selecting agents in the detoxification systems of herbivorous insects [20]. Detoxification and metabolic mechanisms in insects play a leading role in adapting to host defense responses. Insects have evolved various mechanisms to cope with phytochemicals [49]; the ingestion of plant toxins usually results in insect-induced detoxifying enzyme genes. In *Helicoverpa armigera,* cytochrome P450s are induced [53,56] and GST genes are induced in the MED whitefly, *Bemisia tabaci* [57]. Most of the detoxifying enzyme genes in TF6 were significantly downregulated, which may have caused the higher mortality of TF6 in the pre-adult stage due to the adverse effects of ingesting *S. tuberosum* allelochemicals (Table 1) [15]. As the induced defense against plant allelochemicals may take longer than that against single-component pesticides [58], the detoxifying enzyme genes in MF6 were much higher than those in TF6 (Table 5), which may be important for MF6 survival. These results support the hypothesis that *P. marginatus* feeding on *C. papaya* can activate defense mechanisms, which may help them to adapt to the host. In this study, MF6 had higher pre-adult survival rates (Table 2) [15], which is consistent with the fact that cytochrome P450 can improve the survival rate of pesticide-treated citrus fruit flies [59], and the downregulation of detoxification enzymes can lead to the reduced growth of bollworm, *Helicoverpa armigera*, larvae [60] and other species, such as whiteflies [58] and *Chilo suppressalis* [50]. Taken together, these data show that MF6 has a much higher detoxification capacity than TF6. Therefore, *P. marginatus* can successfully detoxify toxic chemicals by feeding on *C. papaya*; higher detoxifying enzymes could play the dominant role. Our findings support the contention that *P. marginatus* feeding on *C. papaya* supports greater adaptability than on *S. tuberosum*, suggesting that *C. papaya* is a more suitable host.

In nature, insects must reproduce to survive. Insect vitellogenesis determines ovarian maturation and ultimately affects insect fecundity. Endocrine hormones, particularly juvenile hormones (JH), regulate the synthesis of vitellogenin (VG) [23]. A lack of nutrients can inhibit the transcription and synthesis of VG, thus hindering the development and maturation of eggs [61]. The C2 protein is primarily involved in regulating cholesterol biosynthesis and metabolism in organisms. Cholesterol is a precursor for hormone biosynthesis acting on signaling pathways [62]. In addition, the gene is involved in the immune signaling pathway [63,64]. The VG gene also plays a role in the innate immune response [65]. We speculate that after feeding on a suitable host, the Niemann–Pick C2 protein gene is upregulated, which regulates the immune signaling pathway, resulting in the VG gene, which is initially used for the innate immune response, being concentrated in reproductive expression and eventually exhibiting strong fecundity. JH is a hormone that can maintain larval characteristics while also promoting adult ovarian development; thus, high JH levels result in longer larval stage, longer development duration, and higher oviposition. When *Spodoptera exigua* larvae were infected with autographa californica multiple nucleopolyhedrovirus (AcMNPV), two hormone-binding proteins were downregulated, both participated in hormone regulation [66]. Thus, Niemann–Pick C2 protein, JH, VG, and VG receptor genes are important in insect reproduction. In this study, six reproduction-related genes were confirmed, all downregulated in TF6 compared to MF6. Combined with the longer oviposition period and higher pre-adult survival rate and fecundity in MF6 (Table 2) [15], these results suggest that these reproduction-related genes may participate in the growth and reproduction of *P. marginatus*.

Ribosomes are another change in *P. marginatus* gene expression to accommodate different hosts, with highly conserved and traditionally considered to have stable gene expression, primarily functions in cell viability, and protein translation. Therefore, ribosomal genes transcript levels have been used to assess stress-induced cellular damage [67,68]. The expressions of five TF6 ribosomal genes changed after feeding on *S. tuberosum*. All five ribosomal genes had been annotated as ribosomal protein genes, which were involved in the synthesis of intracellular proteins, with three significantly downregulated and two upregulated (Table 5; Appendix A in detail). These results show that the process of protein synthesis was significantly influenced in MF6, while it was inhibited in TF6, consistent with MF6′s higher tolerance to *C. papaya*. Recent research found that high concentrations of Butyl Benzyl Phthalate (BBP) significantly downregulated ribosomal RNA transcription on *Chironomus riparius* [68]; therefore, ribosomal gene expression in TF6 was inhibited, most likely due to some plant allelochemicals in *S. tuberosum*. Other studies have reported many insect ribosomal genes to be downregulated, including *Metarhizium anisopliae* [69], *C. suppressalis* [50], and *Anopheles gambiae* [70], in response to unsuitable diets or virus infection. *Paracoccus marginatus* fed *C. papaya* grew faster than those fed *S. tuberosum* (Table 1) [15], indicating that *C. papaya* is a more suitable plant for *P. marginatus*.

Lysosomes, the digestive organs in cells, perform digestion and detoxification functions. Cell autolysis, defense, and the utilization of some substances are linked to lysosome digestion, which is necessary for the body to renew its tissues. In the lysosome pathway, DEGs primarily contained the lysosomal cathepsin B, peptidase C13 family, cathepsin L, papain family cysteine protease, glycosyl hydrolase family 38, glycosyl hydrolase family 31, and trypsin. Lysosomal cathepsin is involved in apoptosis [71,72], and cathepsins can be desorbed into the cytosol and start the apoptosis pathway [73]. Cysteine proteases are commonly found in lysosomes, where they are mainly involved in the phagocytosis, elimination, and digestion of intracellular excess substances. “Lysosome” was an enrichment pathway in the KEGG pathway analysis (Figure 10), with 23 DEGs in TF6 compared to MF6, including 20 downregulated and three upregulated genes. This result is consistent with the “lysosome” pathway in whiteflies, which was downregulated when they were transferred to an unsuitable host [58]. The downregulation of the lysosomal pathway indicates that the detoxification effect was weakened. Lysosomes are protein degradation systems [74] that generate peptides, which are then degraded into shorter amino acid sequences to synthesize new proteins [75,76,77]. In the lysosome pathway, most DEGs were downregulated in *P. marginatus* that were fed *S. tuberosum* compared to those fed *C. papaya*, suggesting that the protein degradation metabolism was regulated. The results show that the ingested *S. tuberosum* sap proteins were ineffective.

Previous research found that the synergistic longevity of *Caenorhabditis elegans* needs DAF-16 (FOXO) positive feedback regulation [78]. The PI3K/Akt/TOR signaling pathway was inactivated because of dietary restriction (DR) in *Drosophila melanogaster*, thereby activating the FOXO signaling pathway participating in survival processes [79]. Compared to MF6, six longevity DEGs were upregulated in TF6 in the “longevity regulating pathway-worm” pathway. We hypothesize that the upregulation of Akt may inhibit the phosphorylation of DAF-16, and the downregulation of DAF-16 shortens the life span of TF6, which is consistent with the study of the life table of *P. marginatus,* which found that TF6 has a shorter lifespan (Table 1) [15].

In addition to host plants, environmental factors or natural genetic variations may influence the gene expression patterns observed by our sampling strategy. Many studies have shown that insects diverged into distinct populations due to their adaptation to different host plants [80,81,82,83,84]. Our findings also indicate that the differences in the life table between MF6 and TF6 were mainly influenced by the host plant [15]. Thus, we believe that the host plant primarily causes the various gene expression patterns noticed in our study.

## 5. Conclusions

We used de novo sequencing to analyze transcriptional responses in this study, providing a preliminary step toward understanding the molecular mechanism by which *P. marginatus* adapts to different host plants. According to our findings, many DEGs were regulated in the host-plant adaptation process of *P. marginatus*, and the majority of DEGs were downregulated in TF6 compared to MF6. The results show that *P. marginatus* fed on *C. papaya* had greater tolerance and adaptability. Our subsequent study will determine whether differential gene expression is related to host plant traits and verify the specific functions of these genes. Therefore, our research offers the first insights into the transcriptional responses of *P. marginatus* when feeding on different host plants.

## Figures and Tables

**Figure 1 insects-13-00850-f001:**
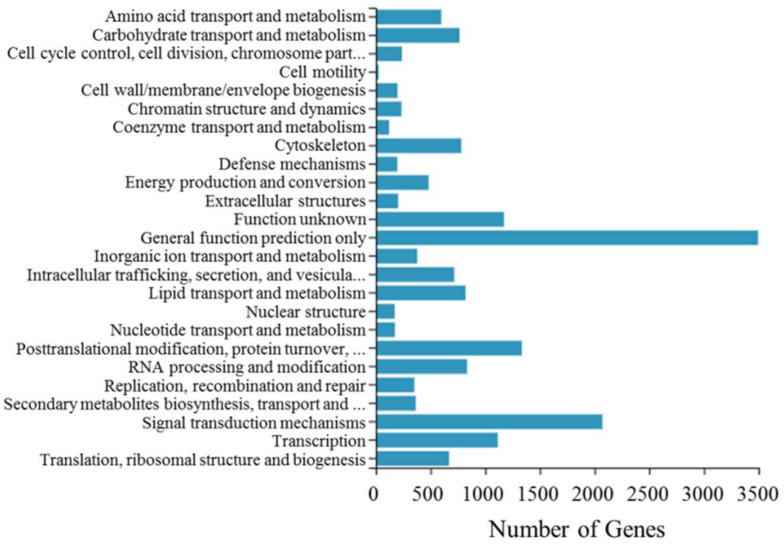
Histogram of EuKaryotic Ortholog Groups classification.

**Figure 2 insects-13-00850-f002:**
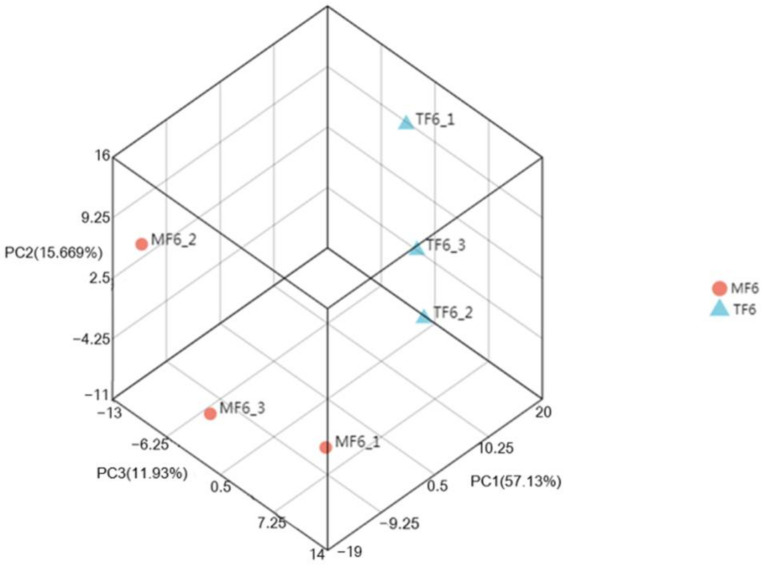
Principal component analysis of the two datasets, MF6 (sixth generation on *C. papaya*) and TF6 (sixth generation on *S. tuberosum*). Each spot represents one sample. Based on transcriptome data of cluster samples. PC: principal component.

**Figure 3 insects-13-00850-f003:**
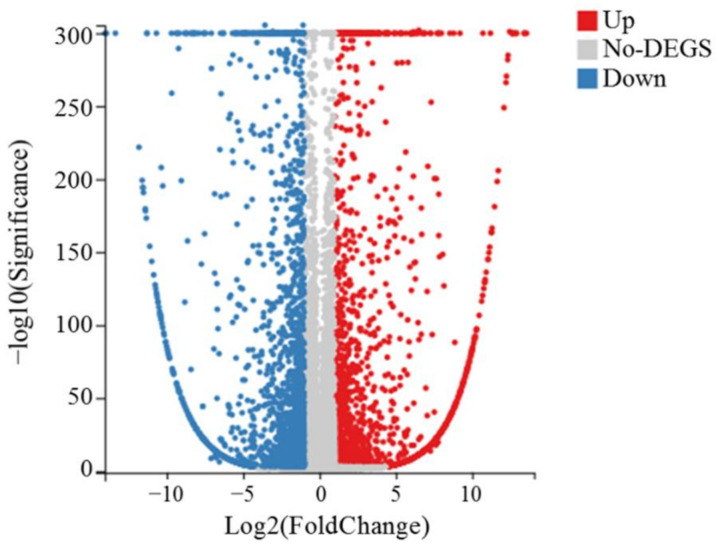
Transcriptional differences between *Paracoccus marginatus* reared on *Carica papaya* (MF6) and *Solanum tuberosum* (TF6). Differentially expressed transcripts are highlighted on the volcano map, with red (2006 unigenes) (q < 0.05, Log_2_ (fold change) >2) and blue (2394 unigenes) (q < 0.05, Log_2_ (fold change) <−2), respectively. Marked in red are transcripts with significant differences in the high expression level, marked in blue are transcripts with significant differences in the low expression level, and marked in gray are transcripts without significant differences.

**Figure 4 insects-13-00850-f004:**
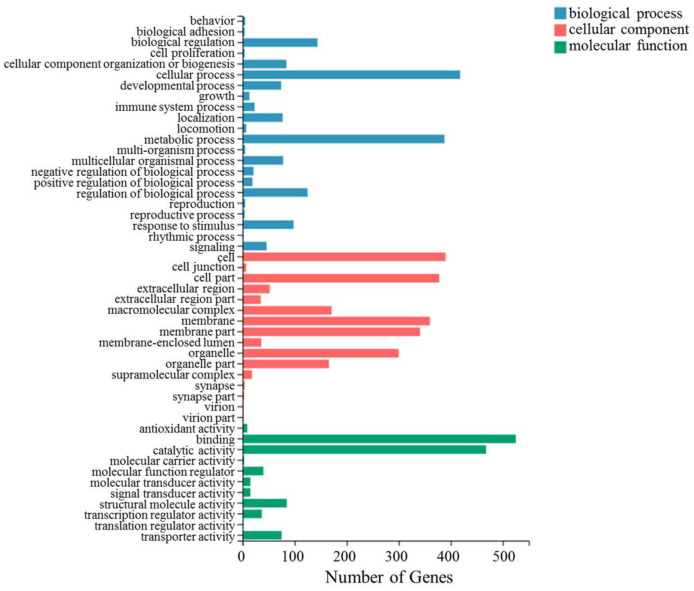
Histogram of Gene Ontology classification.

**Figure 5 insects-13-00850-f005:**
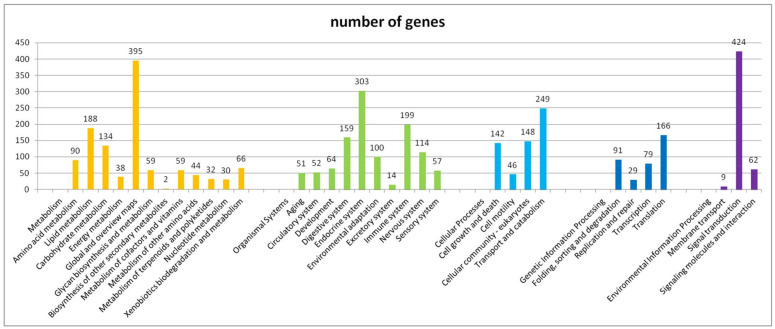
The *Paracoccus marginatus* metabolic pathway as defined by the Kyoto Encyclopedia of Genes and Genomes (KEGG). The x-axis includes the names of the KEGG metabolic pathways, and the y-axis indicates the gene number (level 2).

**Figure 6 insects-13-00850-f006:**
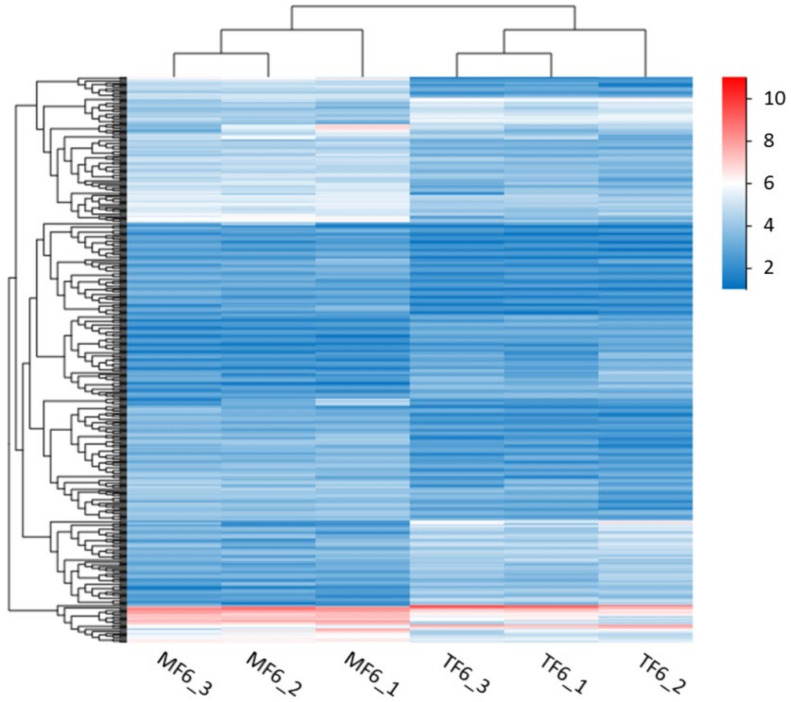
Expression quantity clustering heat map of DEGs based on the fragments per kilobase of exon per million fragments mapped (FPKM) value in MF6 and TF6. The blue strips represent low levels of gene expression, and the red strips represent high levels of gene expression.

**Figure 7 insects-13-00850-f007:**
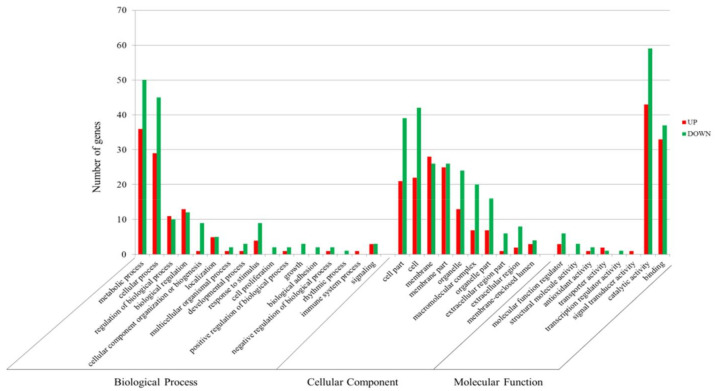
Level 2 Gene Ontology terms of TF6 over MF6.

**Figure 8 insects-13-00850-f008:**
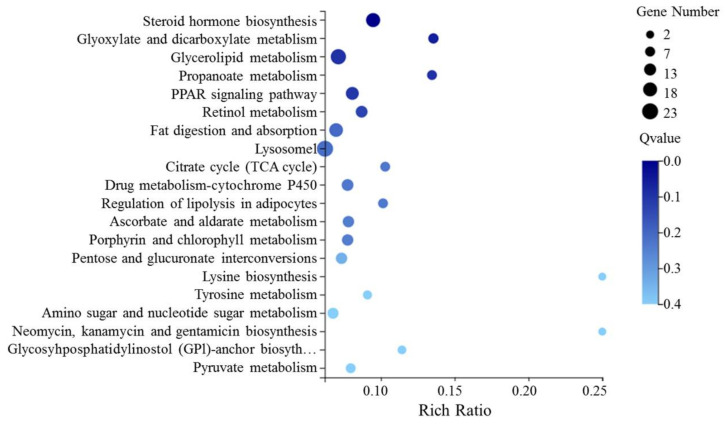
Kyoto Encyclopedia of Genes and Genomes pathways enriched for DEGs.

**Figure 9 insects-13-00850-f009:**
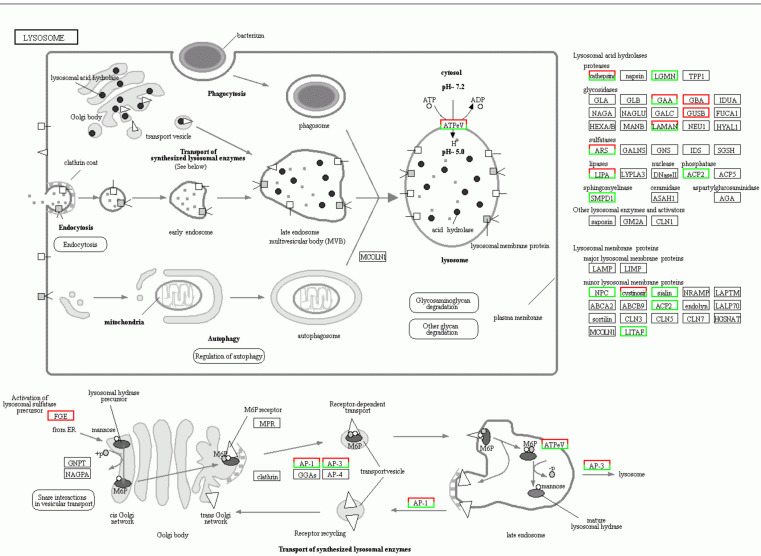
DEGs in the lysosome pathway of *Paracoccus marginatus*.

**Figure 10 insects-13-00850-f010:**
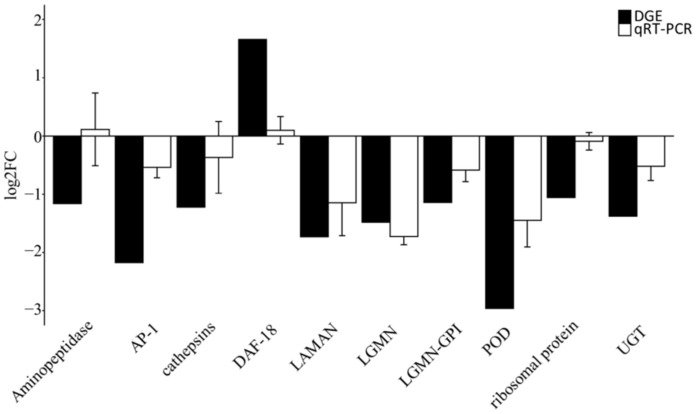
Expression profiles of ten unigenes validated by qRT-PCR. The x axis represents ten unigenes: CL637.Contig13_All, Aminopeptidase; CL1124.Contig3_All, AP-1; CL3981.Contig2_All, cathepsins; CL525.Contig3_All, DAF-18; CL286.Contig13_All, LAMAN; Unigene11195_All, LGMN; CL2961.Contig1_All, LGMN-GPI; CL285.Contig1_All, POD; CL55.Contig2_All, ribosomal protein; CL256.Contig2_All, UGT. The y-axis represents genes’ relative expression levels of genes. Tubulin was used as an internal control.

**Table 1 insects-13-00850-t001:** Developmental stages and adult lifespan of *Paracoccus marginatus* on *Carica papaya* and *Solanum tuberosum*.

Parameters	Development Stage		*C. papaya*		*S. tuberosum*	
			*n*	Mean ± SE	*n*	Mean ± SE
Developmental time (days)	Egg		95	6.21 ± 0.04 ^a^	76	6.17 ± 0.07 ^a^
	First instar		67	5.84 ± 0.13 ^a^	45	5.62 ± 0.09 ^a^
	Second instar		62	5.68 ± 0.16 ^a^	41	5.22 ± 0.17 ^b^
	Third instar	Female	34	3.85 ± 0.10 ^b^	21	5.00 ± 0.44 ^a^
		Male	27	1.59 ± 0.10 ^b^	17	1.88 ± 0.12 ^a^
	Pupa	Male	27	3.74 ± 0.11 ^a^	17	3.94 ± 0.16 ^a^
	Adult	Female	34	28.56 ± 1.84 ^a^	21	18.73 ± 1.11 ^b^
		Male	27	4.11 ± 0.14 ^a^	17	4.41 ± 0.12 ^a^
Total longevity (days)	Pre-adult		39	22.21 ± 0.29 ^a^	62	22.18 ± 0.39 ^a^
	Female		34	49.56 ± 1.83 ^a^	21	40.52 ± 1.04 ^b^
	Male		27	27.85 ± 0.43 ^a^	17	27.12 ± 0.35 ^a^
	All adults		100	28.76 ± 1.77 ^a^	100	19.61 ± 1.30 ^b^
Stage mortality	Pre-adult		39	0.39 ± 0.05 ^b^	62	0.62 ± 0.05 ^a^
	Adult	Female	34	0.34 ± 0.05 ^a^	21	0.21 ± 0.04 ^b^
		Male	27	0.27 ± 0.04 ^a^	17	0.17 ± 0.034 ^a^

Note: Different superscript letters in the same row indicate significant differences by the bootstrap test (*p* < 0.05).

**Table 2 insects-13-00850-t002:** Pre-adult survival rate, oviposition period, and fecundity of *Paracoccus marginatus* on *Carica papaya* and *Solanum tuberosum*.

Population Parameters	*C. papaya*	*S. tuberosum*
	Mean ± SE	Mean ± SE
Pre-adult survival rate	0.61 ± 0.05 ^a^	0.38 ± 0.05 ^b^
Adult pre-oviposition period (days)	4.93 ± 0.19 ^b^	7.44 ± 0.70 ^a^
Total pre-oviposition period (days)	25.87 ± 0.36 ^b^	28.72 ± 0.74 ^a^
Oviposition (days)	15.70 ± 0.70 ^a^	8.39 ± 0.69 ^b^
Fecundity (egg/female)	608.50 ± 52.14 ^a^	177.95 ± 30.00 ^b^

Note: Different superscript letters in the same row indicate significant differences by the bootstrap test (*p* < 0.05).

**Table 3 insects-13-00850-t003:** Transcriptome sequencing data summary from the MF6 and TF6 samples.

Sample	Raw Reads(M)	Clean Reads(M)	Clean Bases (Gb)	Mean Length	N50	Q20 (%)	Q30 (%)	Ratio (%)	GC (%)
MF6_1	72.74	69.65	6.97	1429	2291	97.69	91.26	95.75	39.7
MF6_2	72.74	70.34	7.03	1448	2308	97.91	91.87	96.69	39.58
MF6_3	72.74	69.39	6.94	1475	2342	97.59	90.84	95.39	39.64
TF6_1	75.2	70.73	7.07	1425	2274	97.71	90.38	94.06	39.49
TF6_2	72.17	66.88	6.69	1472	2348	97.41	89.47	92.67	39.4
TF6_3	72.69	67.65	6.76	1432	2308	97.57	89.97	93.06	39.42

**Table 4 insects-13-00850-t004:** Review of functional annotation of the assembled unigenes. †: the nr database, NCBI; ‡: the nucleotide database, NCBI; §: the Kyoto Encyclopedia of Genes and Genomes; ¶: EuKaryotic Orthologous Groups; ß: the Pfam database, European Bioinformatics Institute; ¿: the Gene Ontology (GO).

Values	Total	NR ^†^	NT ^‡^	Swissprot	KEGG ^§^	KOG ^¶^	Pfam ^ß^	GO ^¿^	Intersection	Overall
Number	31,175	20,861	7159	17,880	18,218	17,508	19,614	7464	2029	22,973
Percent	100%	66.92%	22.96%	57.35%	58.44%	56.16%	62.92%	23.94%	6.51%	73.69%

**Table 5 insects-13-00850-t005:** Outline of candidate DEGs associated with digestive enzymes, detoxifying enzymes, reproduction, and ribosomes in the *Paracoccus marginatus* transcriptome.

Category	DEGs	Number of DEGs		
		Total	Down	Up
Digestive enzyme genes	Trypsin	5	5	0
	Cysteine protease	4	4	0
	Amylase	2	2	0
	Carboxypeptidase	1	1	0
	Serine protease	11	9	2
	Lipase	6	3	3
	Aminopeptidase	3	2	1
Detoxifying enzyme genes	Cytochrome P450	4	3	1
	Esterase	14	11	3
	Peroxidase (POD)	3	2	1
	UDP-glycosyltransferase (UGT)	1	1	0
Reproduction related genes	Niemann–Pick C2 protein	1	1	0
	Juvenile hormone	1	1	0
	Vitellogenin	3	3	0
	Vitellogenin receptor	1	1	0
Ribosome	Ribosomal protein	5	3	2

## Data Availability

Raw read sequences can be found under the accession number PRJNA769538 of the NCBI Short Read Archive (SRA) database.

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
