# Peer review of "De Novo Transcription Responses Describe Host-Related Differentiation of Paracoccus marginatus (Hemiptera: Pseudococcidae)"

_insects, 2022, doi:10.3390/insects13090850_

Round 1

Reviewer 1 Report

The manuscript reports transcriptome responses of an important pest species, the papaya mealybug, on two different host plants, papaya and potato. The research provides an important insight on how this papaya pest could evolve to adapt to an alternative host plant. The manuscript utilized a series of conventional methods in analyzing the transcriptome data with 3 replicates. Overall, the methodologies were sound and the discussion went comprehensive and thorough. However, there are still something to be clarified and improved. See my comments below:

Major points

1. Sequence deposit

The transcriptome sequences cannot be found in NCBI. Did you deposit the sequences yet?

2. Statistical analysis

It says that the data were analyzed using TWOSEX-MSChart (Line 169), but it is not sufficient to understand what kind of statistical analysis was used for, especially, Table 1, Table 2 and Figure 11. It seems that mean values were compared between two treatments. I think Student t-test would perfectly work for all these analyses.

3. Table 1

Table 1 should be thoroughly revised. Here are only some examples: Keep a consistent style of fonts in bold or regular. The Development Stage column looks too complicated to understand. Remove the line under Egg, instead add some lines between second instar and third instar; and between pupa and adult. On the Total longevity row, why are there two females? On the same row, “All” under the Development Stage column seems to be a sum of Pre-Adult, Female, and another Female (must be Male). Why don’t you move this “All” row to the bottom of that category? On the Stage mortality row, put “Pre-Adult” in the middle of that section as shown above. Most of all, statistical analysis should be revised. It says in Note, “different lowercase letters in the same column indicate significant differences …”. Did you compare the mean values on the same column? It seems like you compared two mean values between two hosts. What does the bootstrap test mean? I suggest Student t-test. I d

4. Table 2

Again, revise the statistical analysis. You compared means between two host plants, but not among those in the same column. Why did you parenthesize the two parameters: Pre-Adult survival rate and Fertile female ratio? What is the Pre-Adult survival rate? How did you calculate the “rate”? Does it mean the portion of survivals at the state of Pre-Adult? For example, 61% survived on C. papaya, while 38% survived on S. tuberosum? How did you measure Sex ratio and Fertile female ratio? Is the Sex ratio the portion of female to male or male to female? Whatever the sex is, it seems to me that the sex ratio is NOT stable (i.e. not 1 to 1 between male and female), but it says “the sex ratio … is relatively stable …”. I cannot find any description or explanation in Results or Discussion why the sex ratio is biased.

5. Table 3

“Digital Gene Expression” is firstly introduced in the table title, but nowhere else. Provide this full name of DGE in M&Ms (section 2.2).

6. Figure 1

It is obvious that P. marginatus is a member of Hemiptera (Lines 204-20), since it is mentioned on the title. Instead, I suggest you mention that the other 3 species are Hemiptera and discuss why descent number of genes (11.1%) were matched with a termite (Z. nevadensis) genes. Also, more than half genes (53.97%) were matched with “Others”. Explain what the “Others” category includes and why? The letters look blurred, making not easy to read: the figure quality should be enhanced (also in other figures).

7. Figure 3

Add a short explanation in the figure caption what the two datasets, MF6 and TF6, refer to.

8. Figure 4.

“4400 unigenes” is shown with TF6 in the figure caption. It could mislead that the 4400 unigenes are only found in TF6 dataset, but these unigenes are differentially expressed, but found in both two datasets. Also, “On the x-axis, the fold changes represent … on C. papaya and S. tubersum, respectively.” (Lines 235-236) doesn’t sound clear. It shows the up- and down-regulated on C. papaya. If you want to show both plant species, revise it accordingly.

9. Figure 6

Some letters on the far-left side are not shown. Revise it.

10. Figure 8

“MF6-vs-TF6” in the figure caption could mislead that the bars (up or down) are from MF6 over TF6. As far as I understand, the up or down-regulated categories are from TF6 over MF6. Add Y-axis title.

11. Figure 10

Not clearly read. Also, an outer rectangular line is not complete. Either remove all lines or add on top. Provide a higher-resolution version of this figure. The figure caption shows DEG with its full name for the first time after the first introduction in M&Ms (Line 127). But, why only full name in Figure 9 caption? Why only “DEGs” is used in Table 5 without full name?

12. Figure 11

Letters are too small to read.

13. Table 5

Fonts and style: bold or regular, capital or small letters. Number of DEGs would be better than DEGs Number. Abbreviations are used only POD and UGTs: Use them for all the other or not.

14. Discussion needs to be thoroughly revised. References should be more carefully chosen to be relevant to your claims. Below are only some examples, but there are more than them:

Line 369             Proteases degrade peptide bonds found in proteins. Lipase and amylase are not proteases. Revise it accordingly. Also, Reference #36 seems to be out-of-date and too general. Cite a latest one.

Lines 379-380    This sentence doesn’t seem to be relevant in this context. The striped rice borer is a Lepidoptera insect. Is there any other reference relevant to, at least, Hemipter?

Line 383             What do you mean “identical”? The serpin-like gene in this sentence refers to the rice borer (Reference #43). Do you mean that a digestive gene identified from the mealybug is identical to the one from the rice borer? Identical to sequence level?

Lines 384-385    The reference #44 is about a Gram-positive bacteria species. Is it relevant to the corresponding statement?

Lines 390-391    The reference #47 doesn’t seem to be relevant to insect trypsin.

Lines 391-393    Remove this sentence. It doesn’t mean anything.

Lines 410-411    Remove the Reference #20 from this sentence, instead cite more relevant ones from functional studies.

Lines 435-437    The Reference #59 deals with the silkworm infected with a fungal pathogen, which is not a good example to support the claim in one sentence above. What is “hpi” by the way?

Line 471             Ribosomal proteins do not directly refer to “ribosome”. As far as I understand, you identified some ribosome-associated genes differently expressed. Clarify it.

Minor points

Line 207             Remove “Nr annotate species distribution”

Line 232             Remove “Volcano map”

Line 254             Remove this phrase

Line 311             Remove this phrase.

Line 313             Remove an additional period at the end of the caption.

Line 338             cytochrome P450, esterase, peroxidase, and UGT genes

MF6 and TF6     Not a serious issue, but out of curiosity. Why did you name MF6 and TF6? I can guess F6 stands for the sixth generation, but why M for C. papaya and F for S. tubersum. Not-an-easy connection with the abbreviation makes readers confused.

Author Response

Please see the attachment. The red font is the response. Thank you!

Reviewer 2 Report

In methodology, when the host was collected, where they already infested by the insect or cleaned (if yes how?)

Line 89-90: you didnt mention the box before, what was it used for.

how did you acquire the insect and at what stage, these should be stated.

Line 93- 94: when you say "untill death" the information about their number of life span/ generation, reproductive or general biological characteristics should be presented in the introduction. so one can have an idea of how long the data was collected untill death.

Line 94: What experimental data ? please specify because of the word "counted " that was used.

Line 97- 98: Just like i said before, put in your introduction, the duration of each generation and was these generational sampling of the insect or the plant host?

From page 6 -9 and 11, should be placed in the supplementary file. because it drifts the focus of the reader away from the main report, moreover, non molecular experts may not know what to do with the information

Figure 11: axis label should be rewritten to be legible/big/bold enough 

Is Table 5 for P. marginatus in C. papaya's host or the other?

in your molecular analysis, was there a control?

Line 431- 433" the statement seem not to be accurate. higher mortality was observed in host C. papaya in female and male insect except in pre-adult stage, and this is shown in Table 1 and not Table 2.

In Table 1 and 2, result reveals a sex dependent response particularly in terms of developmental duration, longevity and mortality. also emphasis was shown in the reproductive response of the insect in different host. why was this not considered in the sample type for molecular analysis. that is, separating the sexes to show the molecular response. the adult stage was used, but what is the proportion of male to female in the pool of the sample. because in some of the responses of the insects to host, the male had similar response pattern in both host, then the differential expression observed in the growth, digestion and reproduction etc. of the insect from the 2 host could mostly be a contribution from the females, the will eventually affect your conclusion.

Author Response

(The authors gave the same response as above.)

Round 2

Reviewer 1 Report

Line 107             Provide the full name of DGE.

Line 309             As you mentioned in your response, “MF6-vs-TF6” indicates that MF6 was used as control and TF6 was used as treatment”, the figure caption with “MF6-vs-TF6” could mislead that the bars (up or down) are from MF6 over TF6. You may want to change it into “TF6 over MF6” to make it clearly stated.

Line 356             UGT genes

Line 400             gene was

Lines 401-402    According to your new reference #42, a cysteine protease gene was highly up-regulated in the tarnished plant bug, which do NOT accord with the down-regulation in TF6. Revise this discussion accordingly.

Line 404             serine protease genes

Lines 457-459    The host-plant response is supposed to be different from those of fungal infection. I think the B. mori example is not relevant to the case of your MF6/TF6. I suggest you cite another relevant reference to support your claim.

Reference #36   The new reference #36 is not still relevant to insect digestive enzymes. Cite a relevant one.

Table 5 UDP-glycosyltransferase (UGT)

Table S8             UDP-glycosyltransferases (UGTs) / CL1835.Contig1_All / alpha-trehalose-phosphate synthase => Check whether “alpha-trehalose-phosphate synthase” is a UGT or not.

Author Response

(The authors gave the same response as above.)
